# Semi-classical origin of the extreme magnetoresistance in PtSn$_4$

J. Diaz [1], K. Wang[2], J. Straquadine [1], C. Putzke [1,2], Qun Yang[3], Binghai Yan [3], S. L. Bud'ko [4], P. C. Canfield [4] & P. J. W. Moll [1,2] ✉

The so-called "extreme magnetoresistance" (XMR) found in few conductors poses interesting conceptual challenges which address needs in technology. In contrast to the more common XMR in semi-metals, PtSn$_4$ stands out as a rare example of a high carrier density multi-band metal exhibiting XMR, sparking an active debate about its microscopic origin. Here we report a sharp sensitivity of its XMR upon the field angle, with an almost complete collapse only for one specific current and field direction (B//$b$, I//$a$). Corroborated by band-structure calculations, we identify a singular open orbit on one of its Fermi surface sheets as the origin of this collapse. This remarkably switchable XMR resolves the puzzle in PtSn$_4$ as a semi-classical effect of an ultra-pure, compensated carrier metal. It further showcases the importance of Ockham's razor in uncommon magnetotransport phenomena and demonstrates the remarkable physical properties conventional metals can exhibit given they are superbly clean.

In most metals, a magnetic field on the scale of laboratory magnets provides a weak correction to their resistance $R$, commonly quantified via the magnetoresistance $MR(B) = \frac{R(B)-R(0T)}{R(0T)} = \frac{\Delta R}{R(0)}$. When materials show substantial magnetoresistance, it potentially signals physics beyond the ordinary that promises an interesting academic challenge. At the same time, such materials are of central interest to modern technology, enabling applications in magnetic data storage[1], spin-tronics, position sensing, and many more. That technological need inspires the search for materials that exhibit magnetoresistance at levels of practical relevance. The most prominent large-MR solid-state devices act on the spin degree of the electrons: Both the colossal magnetoresistance (CMR)[2] in transition-metal oxides (~20%/T) and the giant magnetoresistance (GMR)[3,4] in ferromagnetic heterostructures (100%/T) exploit the Zeeman coupling of the magnetic field to spin moments.

On the other hand, the orbital effects of static magnetic fields on metals are typically weak. Famously, the free electron gas shows no semi-classical magnetoresistance due to the complete cancellation of the Lorentz force by the induced Hall field. The weak MR experimentally observed in most metals originates from the deviations of the Fermi surface from free electrons or momentum-dependent lifetimes. This physics is well understood by semi-classical models of transport[5].

Yet some materials with extraordinarily large MR arising from orbital effects have been discovered, mostly non-magnetic semi-metals, and the term extreme magnetoresistance (XMR) was coined to differentiate these orbital MR materials from the distinct physics of the CMR and GMR effects. Positive XMR was initially observed in PtSn$_4$, with a MR of up to $5 \times 10^3$ at 14 T and 2 K[6], and later extended to a list of compounds with MR at helium temperatures commonly in the $10^3$–$10^4$ range at 10 T (e.g., WTe$_2$: $10^3$ (Ref. 7), LaSb: $10^4$ (Ref. 8), NbSb$_2$: $10^3$ (Ref. 9), Cd$_3$As$_2$: up to $10^3$ (Ref. 10), MoTe$_2$: 40 (Ref. 11)). In addition to its high value, a key distinction of XMR is its non-saturating growth with magnetic field at any attainable laboratory field scale. It differentiates them from usual metals which can exhibit large MR given they are clean enough. For example, ultra-pure crystals of zinc[12] reach residual resistivity ratios (RRR) $\frac{R(300K)}{R(0K)}$ >50.000, and easily exceed MRs of $10^3$. This semi-classically well-understood large MR is a simple consequence of their small zero-field resistivity and the compression of all orbital effects to the low-field region following Kohler's rule. In stark contrast to boundless growth in XMR materials, however, the

[1]Institute of Materials (IMX), École Polytechnique Fédérale de Lausanne (EPFL), Lausanne, Switzerland. [2]Max Planck Institute for Structure and Dynamics of Matter, Hamburg, Germany. [3]Department of Condensed Matter Physics, Weizmann Institute of Science, Rehovot, Israel. [4]Ames Laboratory U.S. DOE and Department of Physics and Astronomy, Iowa State University, Ames, Iowa, USA. ✉e-mail: philip.moll@mpsd.mpg.de

high MR of such clean materials saturates at rather low fields (in the case of Zn well below 1 T).

The microscopic origin of the XMR behavior remains an active area of research and debate. One key question concerns if XMR materials form a homogeneous group explained by one (or few) universal mechanisms as suggested by their phenomenological similarity. For example, they generally tend to be multi-band semi-metals, hosting electrons and holes at the Fermi level, and a distinct temperature dependence characterized phenomenologically by an apparent "onset" temperature of the MR, which, however, disappears when rescaled on a Kohler plot. Alternatively, given the vast chemical and physical differences between the materials, possibly materials-specific models may be invoked in each case separately. Identifying the microscopics of the MR in diverse sets of materials will be pivotal to answering this question and may allow us to rationally search for candidate materials with enhanced MR.

Various microscopic models to explain such XMR behavior have been brought forward. Electron-hole compensation ($n_e = n_h$) is a semi-classical mechanism to obtain non-saturating MR in two-band metals due to their vanishing Hall fields and the concomitant strong current trajectory modification[13]. In materials with strong spin-orbit coupling, magnetic-field-driven changes of the Fermi surface form another class of mechanisms. These will be particularly effective if a band-structure anomaly such as a van-Hove singularity or a topological band-crossing point exists close to the Fermi surface[14]. Especially in Dirac systems, the loss of time-reversal symmetry in magnetic fields drives a transition into a Weyl material, which has been argued to suppress a topological back-scattering protection[10,15]. Even more exotic ideas are discussed in interacting systems, such as the field-induced formation of excitonic insulators[16,17], proximity to Lifshitz transitions[11], or parallel conductance in topological surface states[18]. Confirming these mechanisms and finding their hierarchy of importance if multiple ones are present is key to advancing our microscopic understanding of XMR.

While most XMR materials are low-carrier density semi-metals, PtSn$_4$ is an interesting example of an XMR metal characterized by a complex, multi-band Fermi surface occupying the entire Brillouin zone and, accordingly, a substantial carrier density. PtSn$_4$ crystallizes in an orthorhombic structure[19] (SG 68, Ccca) with unit cell parameters $a = 0.6418$ nm, $b = 1.1366$ nm and $c = 0.6384$ nm. It forms alternating layers of Sn and monoatomic Pt planes, stacked along the $b$-direction (Fig. 1).

This pure metal is an excellent conductor at low temperatures ($\rho_0 \sim 40 - 50 n\Omega cm$) with RRR of crystals around 1000. At the same time, its MR reaches remarkably large values[6] of $2 - 5 \times 10^3$ at 2 K in 14 T. The MR grows approximately quadratically in field, following

$MR(B) \sim B^{1.8}$, without any sign of saturation at high fields. The temperature dependence of the MR is well captured by Kohler's rule over a large range of fields and temperatures[6], consistent with a semi-classical origin of the MR scaling as $\rho(B,T) = \rho(\omega_c \tau(T))$ (here, $\omega_c = eB/m_{eff}$ denotes the cyclotron frequency, $m_{eff}$ the effective mass and $\tau$ an isotropic carrier lifetime).

Furthermore, this material hosts a topologically non-trivial band structure, and angle-resolved photoemission spectroscopy has uncovered a line of Dirac nodes close to the Fermi level[20]. Hence an XMR origin based on band topology could be hypothesized. Indeed, further non-trivial textures of d-p orbital mixing on one of the Fermi surfaces have been suggested as an important ingredient in the XMR[21]. Given PtSn$_4$ stands out as a highly metallic, high carrier density XMR metal, unveiling its XMR mechanism addresses the question of a potential universal character directly.

## Results

Here, we perform high-precision angle-dependent MR experiments in PtSn$_4$ and report a collapse of the XMR by two orders of magnitude accompanied by a stark contrast in the field-dependence for magnetic fields closely aligned with the crystallographic $b$-direction, and only for currents flowing along the $a$-direction. The field angle sensitively selects between two transport regimes, one of unsaturated XMR with quasi-quadratic field dependence at almost any angle, and one of saturated, low MR in a narrow angle region around $b$. This novel phenomenon demonstrates an unexpected switchability of XMR within one material and allows for the first time to contrast the same material with and without XMR by rotating the field a few degrees away from the $b$-direction. This sensitive behavior constrains applicable theories. We propose a natural explanation in traditional semi-classical magnetotransport on complex Fermi surfaces without the need to invoke more exotic explanations rooted in topology or interaction effects.

Our experimental study is based on precision transport measurements in Focused Ion Beam (FIB) patterned crystals[22]. Beams of typical dimensions ($20 \times 8 \times 3 \mu m^3$) were directly cut from as-grown bulk crystals and mounted onto sapphire or silicon chips featuring a six-point configuration (Fig. 1). Single crystals were grown in Sn as a self-flux, as detailed in Refs. 6,23. These crystals grow as platelets in the crystallographic $a$-$c$ plane (see Fig. 1), thus the morphology follows well the crystal structure. Crystals were aligned using X-ray diffraction (XRD) before the fabrication, allowing us to precisely define the crystallographic direction of the current flow. To avoid surface contamination issues from the impacting ions, the structures were exclusively machined using Xe ions, which were completely desorbed

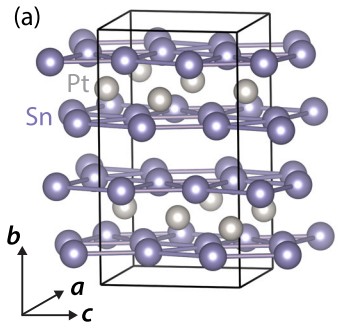
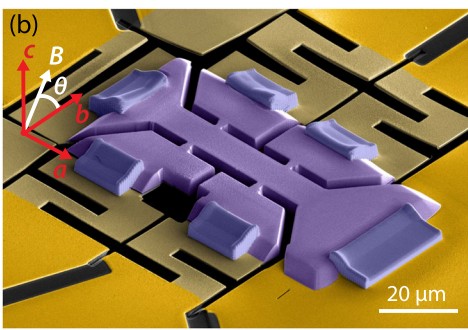
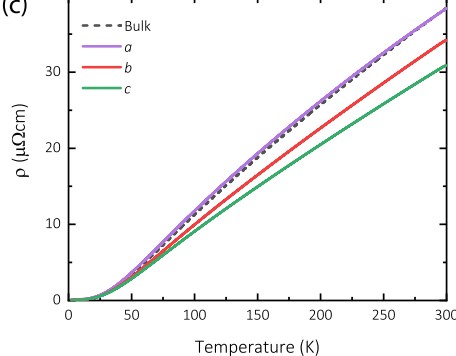

**Fig. 1 | Microstructured PtSn$_4$. a** Orthorhombic crystal structure of PtSn$_4$. **b** Low-stress FIB-prepared crystalline microstructure. The crystal slab (purple) is suspended on a Au-coated SiN membrane (light golden) that has been cut into a spring-shape to minimize transduction of thermal differential contraction. One end (lower right) is anchored to the Si-frame (bright golden) for mechanical stability and rigid angular alignment. The FIB-machined contacts allow for a precise 6-point measurement. **c** Temperature-dependence of resistivity comparing FIB-microstructures along different crystallographic directions and a traditional silver-paint contacted bulk crystal aligned along the $a$-direction.

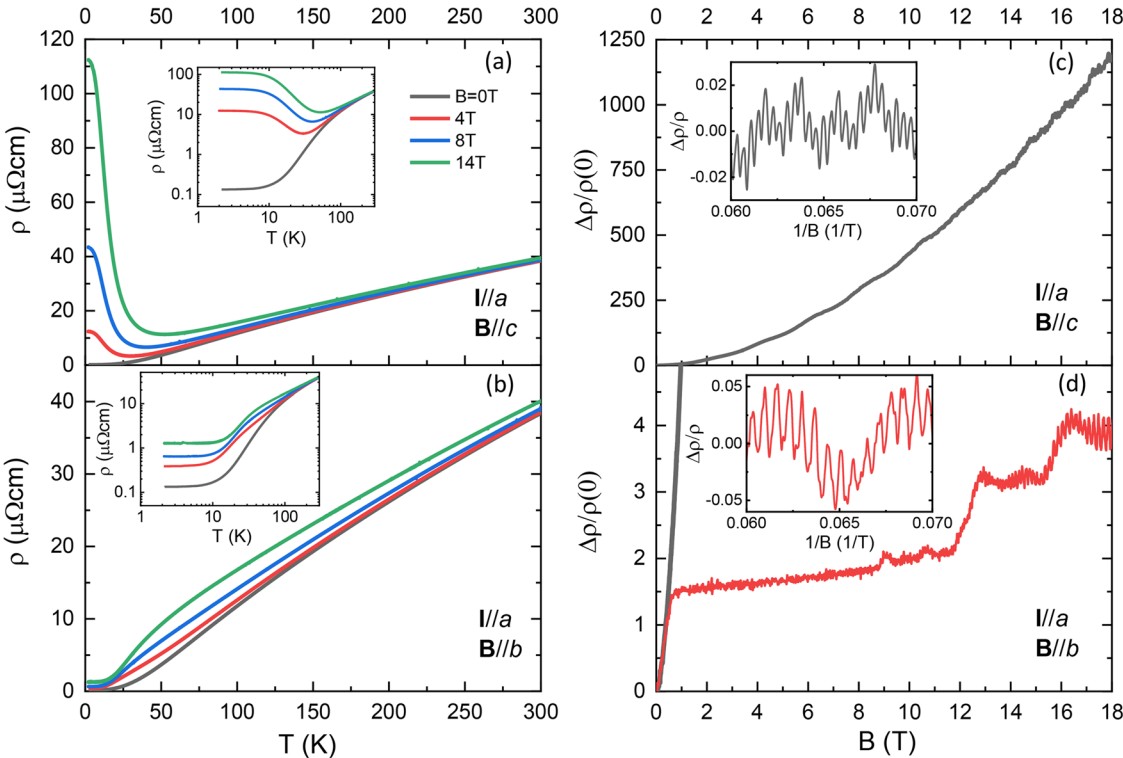

**Fig. 2 | Anisotropy of XMR.** Resistivity of a FIB-microbar of PtSn$_4$ along the crystallographic $a$-direction. While fields along the $c$-direction show a typical XMR behavior (**a**), fields precisely along the $b$-direction (**b**) show a remarkably different metallic behavior. (**c**) For B//$c$, the magnetoresistance at 2 K varies quadratically with field; while (**d**) for the special B//$b$ configuration the initial rapid rise saturates (B//$c$ shown for comparison on the same scale, gray line).

from the material. Coarse cutting was performed at a current of 60 nA, and edges were fine-smoothened using 15 nA. Samples were mounted both straight onto a chip or onto a suspended membrane to check for effects of mismatch strain between the sample and the substrate, which were not observed. For the membrane mounting, the microstructures were transferred in situ onto a 100 μm x 100 μm SiN membrane. The 100 nm thick membrane was gold coated to form the electric contacts (200 nm thick). Using a beam current of 4 nA, the SiN membrane was cut to define suitable electrical contacts that extend to the supportive Si frame. This allows to form solid electric contacts at minimal substrate mismatch strain[24].

The temperature dependences of the microstructures along different crystallographic directions are quantitatively consistent with our own measurements on bulk crystals as well as published resistivities[6] (Fig. 1c). This already hints at an important piece of the physics of PtSn$_4$. Despite its platelike crystal morphology and its structural arrangements into Pt planes, its physical behavior is that of an almost isotropic, strongly three-dimensional intermetallic metal. Similarly, also the magnetic properties are relatively isotropic[6], with a diamagnetic susceptibility ratio as low as $\frac{\chi_{ac}}{\chi_b} \sim 2$.

This transport data, in combination with the clear quantum oscillations, further evidences that the bulk material quality was not drastically changed by the fabrication. Yet microfabrication necessarily alters the transport physics of any material. The direct contribution of the FIB amorphization and damage layer (~20 nm thick, 1.6% of total cross-section) is negligible given the high conductivity of PtSn$_4$, yet finite size effects inevitably occur when the wire cross-section is substantially smaller than the mean-free-path, as is here the case. This leads to well-known semi-classical corrections from enhanced boundary scattering to the total conductance[25,26]. Elastic sidewall scattering increases the resistance of finite size objects above the bulk value following Matthiessen's rule, lowering the RRR to around 300 in these microstructures. In high fields, however, the cyclotron radius

$r_c \sim 1/B$ shrinks sufficiently below the device size, restoring bulk behavior even in micron-sized conductors. The key findings presented here were robustly observed in microstructures as well as bulk crystals alike, albeit much crisper in the microstructures (see supplement).

The temperature dependence of the magnetoresistance in the B//$c$ configuration shows clear XMR below 40 K (Fig. 2), reaching high MR values of $8.7 \times 10^2$ at 2 K and 14 T without signs of saturation. These high values are slightly lower than those of bulk crystals, owing to the finite-size enhanced zero-field resistance entering the denominator. A similarly high MR with a slightly crystalline anisotropy-dependent prefactor is observed at any field direction but one.

Only for fields precisely aligned with B//$b$, a completely different situation unfolds. An initially increasing MR at elevated temperatures terminates at 50 K and collapses at low temperatures. The temperature scale of the collapse coincides with that of the sudden growth along the other configurations. While the MR of around 10 in this configuration remains sizable on an absolute scale, it is two orders of magnitude smaller than in the B//$c$ configuration. In addition to this quantitative difference also the qualitative trends of the functional forms change. Notably, the slope $\frac{d\rho}{dT}$ remains positive at all temperatures, unlike the extended region of negative slope associated with the large increase of MR in the case of B//$c$. This behavior is further unique to currents flowing along the $a$-direction. A similar microbar along the $c$-direction shows regular MR behavior for all field configurations, including B//$b$ (see supplement).

The loss of XMR behavior for B//$b$ is directly visible in the field dependence of the MR itself (Fig. 2c, d). At 2 K, an almost linear initial increase saturates around 0.75 T, followed by a much slower increase up to higher fields. At 12 T and 16 T, a step-like increase followed by shallow plateaus is observed. This behavior occurs in a well-quantized regime and may correspond either to low-frequency quantum oscillations or to quantum limit effects in some of the very small pockets of PtSn$_4$. On top of this signal, high-frequency oscillations appear at high

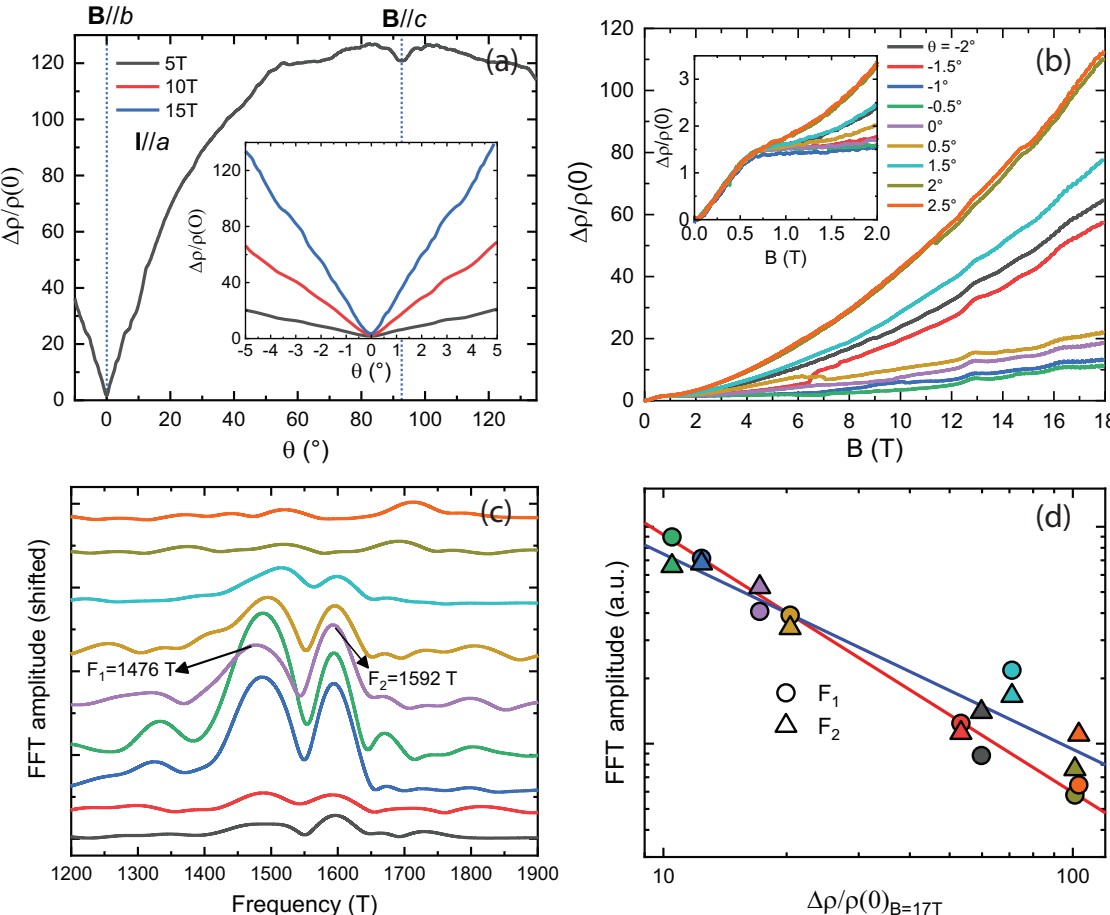

**Fig. 3 | Sharp disappearance of XMR with field-angle. a** Angle-dependence of MR in a $(bc)$-plane rotation at 2 K. The MR collapses sharply for fields well aligned to B//$b$ ($\theta = 0°$) only. **b** MR for various small angles around B//$b$ highlights the sharpness of the effect. Sub 0.1° inadvertent motion of the rotation mechanism during the field sweep leads to the appearance of step-like artifacts in the data. Inset: The low-field range ($\omega_c \tau < 1$) is rather angle-independent. **c** Quantum oscillation spectrum in the 1200-1900T frequency range of the data in **b**. The frequencies F1 and F2 only appear in the narrow region of collapsed MR, indicating this region coincides with a crystallographic high-symmetry direction. **d** A clear correlation between the FFT amplitude of these frequencies and the value of the high-field MR is observed. Datapoints follow the same angular color code as in **b, c**.

fields. For a direct comparison in magnitude, the field dependence of B//$c$ is overlaid in the graph (Fig. 2d, gray line). The B//$c$ MR immediately leaves the relevant scale for the B//$b$ MR owing to the extreme magnitude of its prefactor characteristic for XMR.

A rotation in the $(bc)$ plane highlights the sharp MR minimum for fields aligned with the $b$-direction (Fig. 3a). Tilting the field only 1° away from the $b$-direction dramatically increases the MR by a factor of 8 at 14 T, and further tilting explodes the values towards their XMR levels. To achieve precise field alignment with B//$b$, we utilized high-precision two-axis rotation under a static magnetic field (rotator type: attocube 3DR).

This provides clear evidence that the XMR within one sample of PtSn$_4$ can be switched off by fine-tuning the magnetic field direction, and the field dependence further confirms a quantitatively and qualitatively different transport regime for B//$b$ (Fig. 3). The sharpness of this feature is remarkable, in fact, such measurements pose an experimental challenge. Small jump-like motion of the rotator (<<0.1°) shows up as artificial resistance jumps. This sensitivity is highlighted in Fig. 3b, which shows the field-dependence in 0.5° steps around B//$b$. At these comparatively coarse steps, the true B//$b$ is missed, yet it provides a good overview of the general behavior. The initial slope is field-angle independent, and soon after, the quickly rising quadratic growth sets in. Only at well-aligned fields the curves are flat and dominated by the linear terms. The possibility of eliminating XMR within one

compound by rotating the field a few degrees is a key observation presented here. It sets PtSn$_4$ apart from other metallic XMR materials such as ReO$_3$, which commonly exhibit anisotropic yet always finite XMR in all field directions[27,28].

This phenomenology explains why this feature had been missed in previous studies[6]. First, sub-degree fine-tuning is required to align the field precisely along the $b$-direction. Further, this collapse is only observed for currents flowing along the $a$-direction, while those along $c$ still show strong quadratic growth (see supplement). Without the precise channel fabrication of the FIB process, measurements on bulk crystals inevitably mix the resistivity tensor components perpendicular to the field due to unavoidable current injection inhomogeneities in pure metals[29]. This mixing yields an effective average and hence smears out the singular behavior observed only for (B//$b$, I//$a$). The MR in this narrow region also exhibits a further peculiarity in the quantum oscillation spectrum. While the overall frequency spectrum is very complex, reflecting the complex shape of the multiple Fermi surfaces consistent with previous reports[6,30–32], one set of frequencies stands out at B//$b$ with an anomalously enhanced amplitude. This strong oscillation is well visible already in the raw data (Fig. 2). These turn out to be two close-by beating frequencies, $F_1 \sim 1480T$ and $F_2 \sim 1590T$. These anomalously strong frequencies are only detected in the same narrow-angle range around B//$b$ in which the XMR collapsed. This connection can be clearly visualized by the direct correlation between

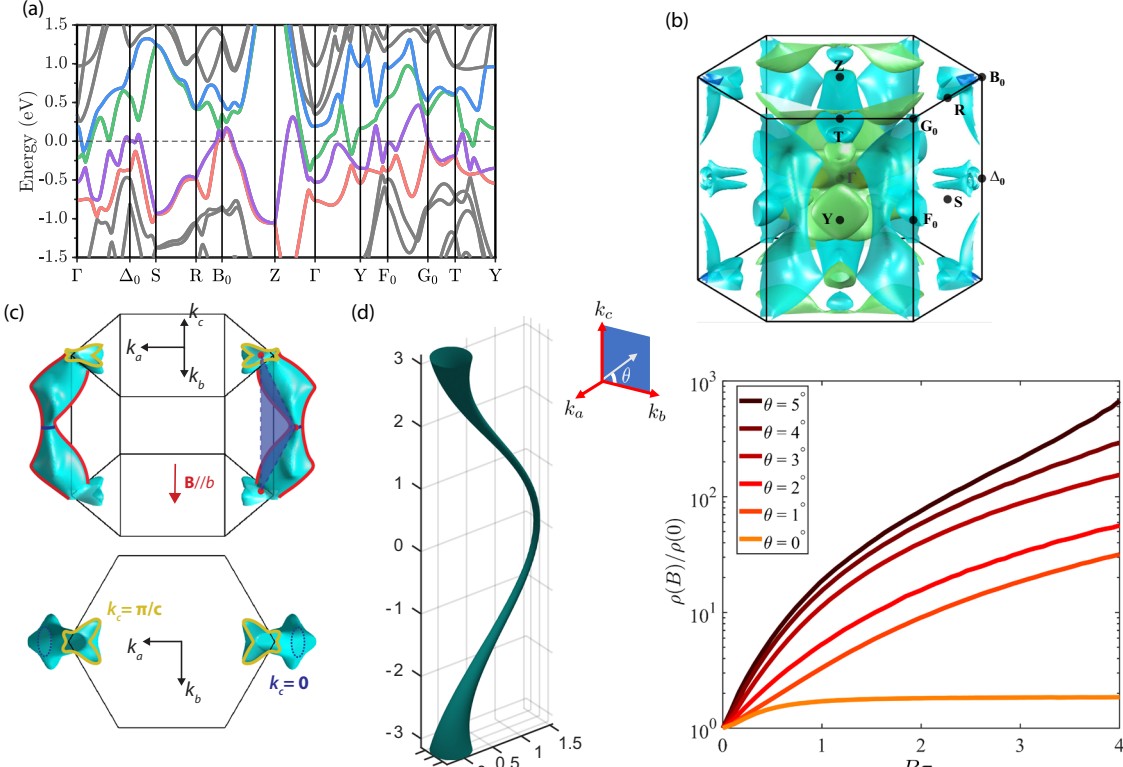

**Fig. 4 | Microscopics of the vanishing XMR.** Band-structure (**a**) and Fermi surface (**b**) from first-principles calculation. Multiple electron and hole bands cross the Fermi level, resulting in a multitude of complex Fermi surface sheets. The angle selectivity arises from a single open orbit, highlighted in an extended zone scheme in **c**. The narrow necks in the $k_c = \pi$ and $k_c = 0$ planes (red dots) act as filters, allowing only open orbits when the field is precisely aligned perpendicular to the surface formed by them, the crystallographic $b$-direction. **d** Cylindrical Fermi surface of the transport model featuring a displaced neck emulating the direction filtering features of the more complex PtSn$_4$ Fermi surface. The calculated MR on this model resembles the experimental data well. It shows a significant increase of MR upon slight deviation from $\theta = 0°$, at which the MR saturates due to the open orbit. $\tau$ denotes the scattering time.

the oscillatory amplitude and the magnitude of the MR (Fig. 3). As the MR quickly increases with increasing field angle, these oscillations fade away. This correlation indicates that, indeed, the MR anomaly occurs for fields along a symmetry direction of the Fermi surface and not a direction close to but distinct from $b$.

To understand the microscopics of these transport signatures, we first turn to band structure calculations. The electronic structure calculations were performed within density functional theory as implemented in the Vienna ab initio simulation package[33,34]. The exchange-correlation interaction was described by the generalized gradient approximation as parametrized by Perdew-Burke Ernzerhof[35]. The cutoff energy was set to 350 eV for the plane wave basis. We projected the Bloch wavefunctions into Wannier functions[36], and constructed the tight-binding model Hamiltonian based on Wannier functions. The Fermi surfaces were then calculated with a $120 \times 120 \times 120$ k-grid from the tight-binding model Hamiltonian.

The resultant band structure shows a highly dispersive, multi-band metal hosting many geometrically complex Fermi surfaces, in agreement with previous reports[6]. This Fermi surface is obviously electronically highly three-dimensional, and no significant anisotropy is to be expected, as observed. This 3D behavior reflects the participation of both Sn and Pt in metallic transport despite the layered crystal structure. Various avoided band crossings fall close to the chemical potential, which cause small electron-like and hole-like pockets at low symmetry points, as has been noted previously[30,32]. In addition to the various small pockets, three larger sheets exist. At the $\Gamma$ point, a large central sheet with a highly complex internal structure appears. A second one is an elongated pocket parallel to the $\Gamma - Y$ line (commonly referred to as "sausage-shaped", blue in Fig. 4). This pocket

is the remanence of a quasi-2D system formed by the structural Pt and Sn layers. It forms a weakly dispersive cylinder along the $b$-direction, which is normal to the layers. Within the crystal structure, however, hybridization opens a significant gap, cutting the cylinder into disconnected sausage-shaped segments. The dimensionality of this sheet, therefore, is also 3D, albeit with an anisotropic, elongated shape.

The third large Fermi surface is the most interesting from a magnetotransport perspective, and its peculiarities are at the origin of the XMR collapse (Fig. 4c). Along the $\Delta_0 - B_0$ line, a corrugated cylinder spans the entire Brillouin zone along the $c$-direction. This sheet is easy to miss in a reduced zone scheme, hence we here complete it in the extended zones (Fig. 4c). Despite its significant warping, this cylinder touches the zone boundary and hence is a sheet of 2D topology. This shows an interesting peculiarity of PtSn$_4$. It does host a 2D electron system after all, however, in the $(ab)$-plane, perpendicular to the structural Pt layers. The relevance of the $b$-direction, when MR is concerned, is not that of an out-of-plane field regarding the Pt layers but that of an in-plane field for this hidden 2D electron system.

This sheet naturally is of key importance for the magnetotransport as it hosts the only open orbit in the system (marked red in Fig. 4). While the topology of this sheet is 2D, the 3D nature of the material is reflected in its strong corrugation. It is deformed significantly such that its cross-section in the $k_c = 0$ plane does not overlap with its cross-section in the $k_c = \pi/c$ plane. Furthermore, around these planes narrow necks form which take the role of orbital filters. Merely a small number of trajectories passes through such a constriction, and to form an open orbit, it must pass through all such points in reciprocal space. In PtSn$_4$, these constrictions define a plane perpendicular to the $b$-direction. The narrowness of the open orbit

anomaly is a direct consequence of this geometric orbital filtering. The role of such constriction networks is well-established in elemental metals and at the heart of the highly complex angle-dependence of the MR of copper[37,38].

Lifshitz, Azbel and Kaganov[13] in 1957 famously demonstrated that the semi-classical transport in the high-field limit $\omega_c\tau\gg1$ is a geometric property of the Fermi surface, such as the saturation of the MR when all accessible electron orbits are closed. This intuitive result is based on the orbital averaging of the velocity to zero in closed orbits. This averaging is naturally absent in open orbits, and a non-saturating MR appears when open orbits are driven. As a result, a sharp peak in the MR appears, like in single crystal copper[39], yet unlike the dip in PtSn$_4$. The non-saturating MR at open orbit conditions, however, comes with an implicit yet critical assumption that there is a dominant carrier type. LAK had shown that compensated semi-metals ($n_e = n_h$) display the exact opposite behavior due to the absence of a Hall field in this case. As a consequence, the MR grows without bounds quadratically in the field when all orbits are closed, opposite to the saturating MR of uncompensated metals. Likewise, when fields drive open orbits, a sharp minimum occurs. The semi-metal zinc in the magnetic breakdown regime is an exemplary case for such a situation[12] and its MR shares striking phenomenological similarities to PtSn$_4$, in particular, sharp minima of saturated MR that otherwise grows quadratically in field. Unlike PtSn$_4$, however there interband tunneling dominates the field dependence of the conductivity tensor.

These observations present a compelling case for an entirely semi-classical transport scenario in an electron-hole compensated semi-metal. In principle, one can integrate Boltzmann equation on the Fermi surface to obtain an ab initio prediction of the resistivity tensor on the semi-classical level[5]. In practice, however, multiple Fermi surface sheets with such complex shapes as in PtSn$_4$ rarely yield a satisfactory quantitative agreement. We note that the open-orbit cylinder cannot be treated in isolation as the key component to the XMR switching behavior lies in the charge carrier compensation. The full Fermi surface needs to be considered in all complexity to restore this compensation in a calculation. Here, the topological argument by LAK provides an elegant geometric solution. When all orbits are closed, the semi-classical MR of compensated metals grows without bounds as $B^2$. Further when open orbits are present, their MR follows $B^2$ for any current configuration but one. One special case arises when the current exactly flows along the direction of non-vanishing Fermi velocity when averaged over one cyclotron period. In this direction, the magnetic field effects cancel, and no MR remains – leading to saturating behavior. This exactly describes the MR observed in PtSn$_4$. In particular, the open orbit extends along the $c$-direction, and for fields along $b$, the Fermi velocity of the open orbit is along $a$. This explains naturally why (B//$b$, I//$a$) shows saturating MR, unlike any other field-current configuration in this compound. To further corroborate this topological result, we perform a semiclassical Boltzmann transport calculation using a simplified model Fermi surface to emulate the displaced narrow neck characteristic observed in PtSn$_4$ (see supplement). The model consists of a narrow cylindrical Fermi surface that is displaced in the $k_c = 0$ plane to capture the orbital filtering mechanism. For clarity, only one cylinder is shown while carrier compensation and time-reversal symmetry necessitate further copies of it for consistency (Fig. 4d). This simplified model captures the data well: A pronounced minimum and field-saturation occur when the magnetic field aligns with the plane defined by its open orbit. This minimum is notably sharp, as even a slight deviation in angle leads to a substantial change in MR when the orbits are closed. It is gratifying to see that such a model captures the essential physics so well, a natural result of the topological nature of the LAK theory.

The broader implications of identifying the semi-classical origin of the transport in PtSn$_4$ are twofold. First, it identifies the system as an interesting testbed to probe quasi-one-dimensional electron transport in a compensated electron-hole sea. In the open orbit configuration, all states are localized perpendicular to the field in Landau levels but one, the open orbit. These states propagate in only one direction, along the $a$-direction, and carry all electric currents. Whether in this regime band-structure topology is important, either on the open-orbit sheet or the auxiliary Fermi surfaces, has not been addressed yet. Identifying the origin of the step-like features apparent in the MR only for this special configuration is a natural challenge. It further shows that semi-classical transport can induce XMR even in complex metals without the need for more exotic theories. The presence of singular open orbits provides a unique possibility to selectively switch XMR on and off, a rather rare material property that further shines light on the microscopic details of charge transport in the XMR state. In light of these results, more investigations into the role of semi-classics in other XMR materials are clearly called for.

## Data availability
The data generated in this study have been deposited in the Zenodo database under the accession code https://doi.org/10.5281/zenodo.11067142.

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

## Acknowledgements
This project was supported by the European Research Council (ERC) under the European Union's Horizon 2020 research and innovation program (JD, CP, PJWM, grant no. 715730). Work done at Ames National Laboratory (SLB, PCC) was supported by the U.S. Department of Energy, Office of Basic Energy Science, Division of Materials Sciences and Engineering. Ames National Laboratory is operated for the U.S. Department of Energy by Iowa State University under Contract No. DE-AC02-07CH11358.

## Author contributions
J.D., J.S., and C.P. performed the microfabrication and magnetotransport experiments. S.L.B. and P.C.C. synthesized and characterized the single crystals. Q.Y. and B.Y. performed the DFT calculations and analyzed the orbits. K. Wang designed and solved the analytical model. S.L.B., P.C.C., and P.J.W.M. designed the experiment. All authors contributed to writing the manuscript.

## Funding

## Competing interests
The authors declare no competing interests.
