## [Peer Review File · Nature Communications]

Reviewers' Comments:

Reviewer #1:

Remarks to the Author:

The manuscript "Semi-classical origin of the extreme magnetoresistance in PtSn4" experimentally investigates the transport properties of the semimetal PtSn4 in a magnetic field. It is found that for certain orientations of magnetic field and applied current, the resistivity becomes extremely high. A simple, quasiclassical explanation of this phenomenon that involves only the structure of the Fermi surface rather than nontrivial topology and the formation of Dirac nodes is provided and linked to the literature (see Ref. [12]). Despite the complicated Fermi surface, the authors were able to identify the presence of an embedded 2D electron subsystem. The observed extreme magnetoresistance originates from the narrowness of the corresponding open orbit, which agrees with the theory by Lifshitz, Azbel, and Kaganov.

The manuscript is well-written, technically sound, uses well-established methods (I cannot comment on the validity of experimental data and sample preparation, however), and addresses an interesting problem. On the other hand, because of the previous findings related to extreme magnetoresistivity, it is not clear how significant are the obtained results (see also a detailed concern below). In addition, the manuscript lacks analytical analysis beyond citing qualitative arguments from Ref.[12].

Thus, before this manuscript can be recommended for publication in Nature Communications, the authors should address the following concerns/suggestions:

1. The manuscript will strongly benefit from a more detailed analytical analysis. I suggest adding a section to the supplemental material discussing the behavior of the resistivity tensor with a magnetic field and, perhaps, temperature. The latter can be important since it governs the deviation from the $n_e=n_h$ limit. In addition, fittings of the experimental data with phenomenological formulas may be also useful in establishing the connection to the theory by Lifshitz, Azbel, and Kaganov.
2. In the introduction, it is stated that the extreme magnetoresistance was also observed in other semimetals such as Cd3As2. The latter, however, is a Dirac semimetal. Is there any 2D electron subsystem there that would allow for an open orbit and a similar explanation? If not, how generic is the proposed mechanism for other materials that show the extreme magnetoresistance?
3. Is there any intuitive physical explanation for why the physical behavior of PtSn4 is that of an almost isotropic, strongly three-dimensional intermetallic metal? This is an interesting observation especially taking into account the layered structure of PtSn4.
4. While the authors try to provide a connection to applications, extreme sensitivity to the direction of the magnetic field may negate all potential gains from the extreme magnetoresistance. The authors should either tune down these discussions or strengthen them, and explain how the observed phenomenon may be used.
5. It is mentioned in the summary that PtSn4 is "an interesting testbed to probe quasi-one-dimensional electron transport in a compensated electron-hole sea". However, the main finding originates from a hidden 2D electron subsystem. Do the authors understand quasi-1D in the sense that the extreme magnetoresistance is observed only along a certain crystallographic axis? This statement should be clarified.
6. The axes in Fig.1a) seem to be wrong and contradict the description in the main text and literature, see also Fig. 1 in E. Mun, H. Ko, G. J. Miller, et al., Phys. Rev. B 85, 035135 (2012). Indeed, while it is stated that "alternating layers of Sn and monoatomic Pt planes [are] stacked along the b-direction", the layers in the figure are stacked along the c-direction. I suggest to double-check the labeling of the axes along the text and to bring them in agreement with the literature.

Reviewer #3:

Remarks to the Author:

The manuscript by Diaz et al. reports an angle-dependent magnetoresistance study on microstructured devices of high purity metal PtSn4. The key finding is an abrupt disappearance of the extreme magnetoresistance (XMR) in the system with magnetic fields along b, and the authors highlighted the similarity of the observation with that known in compensated metal zinc, and used

this to support that the XMR in the system is likely semiclassical and attributed to compensation of the high mobility carriers. The experimental observation and conclusion themselves are robust and the sharpness of the observation is uniquely enabled by fabrication of mesoscopic devices. However, the novelty and significance of the study, beyond elucidating the mechanism of XMR in the system, remain somewhat unclear. Similar physics and phenomenology of compensated metals have been known to elemental Zn, and more recently reported in high mobility ReO₃ with comparable magnitude of magnetoresistance (see e.g. Phys. Rev. Materials. 5, 105004, 2021).

On top of this, one aspect of the manuscript that needs correction is the labeling of the crystal structure as well as crystallographic orientations. Fig.1a displays a crystal structure seemingly containing more Pt atoms than Sn, contradicting the chemical formula. The choice of a, b and c axes in Fig. 1a is inconsistent with most literature on PtSn₄ (e.g. PhysRevB.103.085125 Fig. 1d) and nor is it consistent with the cell parameters cited in page 2 paragraph 3. These conflicting conventions are again used in a mixed manner in the discussion of the DFT Fermi surface and cyclotron orbits. Given the key importance of the relative orientation of magnetic field to the crystallographic axes in the study, it is important to use a notation that does not cause confusion.

Below are a few additional questions:

(a) How robust is the narrowly connected Fermi surface feature in DFT (Fig. 4c) with respect to shifts in Fermi energy or relative band positions, which could occur in real materials?

(b) Does the highlighted quantum oscillation in Fig. 3 correspond to any of the Fermi surfaces shown in Fig.4? Are there any quantum oscillation evidences for the evolution between closed and open cyclotron orbits?

(c) P7 paragraph 1, "Gamma-X" line is mentioned, but no "X" point can be found in Fig.4.

(d) What's the orientation of current in the bulk curve in Fig. 1c?

Dear Reviewers,

We thank you for your time and effort to review our manuscript “Semi-classical origin of the extreme magnetoresistance in PtSn₄”. It is highly gratifying to see that our focus on clarity is remarked by both reviewers in unison. Indeed, it was a focus of ours to make the text as accessible as possible, avoiding field-specific lingo and explaining clearly the physics of this system. This is regrettably missing in many cases, especially when transport properties in topological systems are treated. Before addressing the points one-by-one, we would like to make two overarching points:

Switching XMR off within a material had never been reported. While they clearly share underlying orbital effects as we explain, it is important to note that the physics of the mentioned examples of ReO₃ and Zn is very different. ReO₃ (PRM 5, 105004, 2021) hosts a network of interconnected Fermi surfaces that span the entire reciprocal space, not unlike Ag or Cu. As such, a geometric analysis following Chambers and Shockley reasonably explains the factor of 4 increase of the transverse magnetoresistance observed within a few degrees of field angle (compared to our factor of 120). Zinc physics is very rich and its high-field properties have been shown by Stark (e.g. R.W. Stark, Phys. Rev. A 133, 1698 (1964), see figure below) to arise from magnetic breakdown – a field induced orbit reconfiguration that strongly and non-linearly changes the magnetoresistance. The sharp angle-dependence arises from the exponential sensitivity of tunneling to the field-tuned gap. Enabled by the smallness of the spin-orbit gap in Zn, this limit does not at all apply to PtSn₄.

This touches on a key point: XMR physics is decoupled from the sheer value of magnetoresistance. One can easily obtain enormous values in any metal provided the crystals can be made very clean, from trivial semi-classical reasons. The distinguishing feature of the XMR class of materials is the non-saturating field-behavior of the MR, which sparks the search for more exotic explanations. This can be well observed in the given example of Zn. While its MR is very large, it saturates at very low fields. This behavior is a result of Kohler scaling ($\omega_c \tau$) which compresses all possible MR into the observable field range. This large $\Delta\rho$ is further numerically amplified by dividing it by the small residual resistivity typical for ultra-clean metals.

Zn is a special metal with magnetic breakdown at very low fields dominating the field-dependence of the resistance. Nonetheless, its large MR does saturate as it should, given its imperfect carrier compensation. While very interesting, its well understood physics clearly is distinct from that of XMR.

ReO₃, on the other hand, only exhibits a mild anisotropy in its XMR as a function of field angle (Q. Chen et al., PRB 104, 115104 (2021)) which is explained by orbital network theory. Crucially, it exhibits XMR in any field direction, as defined by its non-saturating growth. Its XMR can be somewhat reduced by field rotation, but it cannot be eliminated given its Fermi surface.

It marks a critical advance in demystifying XMR. The nature and origin of XMR is a vibrant and active debate in our community, to which our observation is an important contribution. PtSn₄ is a special compound and the only one known (so far) in which XMR can be switched off. Hand in hand with a solid explanation, the paper demonstrates how XMR can be constructed based on known transport physics without exotic explanations. While exotic theories are exciting and push the frontier of physics forward, more cautious works explaining unusual data within classical frameworks are often underrepresented in the debate. In reviewing the revised manuscript, we thus hope that our manuscript does not fall into this gap.

With these general considerations, we now discuss the responses point-by-point:

Reviewer #1 (Remarks to the Author):

The manuscript “Semi-classical origin of the extreme magnetoresistance in PtSn₄” experimentally investigates the transport properties of the semimetal PtSn₄ in a magnetic field. It is found that for certain orientations of magnetic field and applied current, the resistivity becomes extremely high. A simple, quasiclassical explanation of this phenomenon that involves only the structure of the Fermi surface rather than nontrivial topology and the formation of Dirac nodes is provided and linked to the literature (see Ref. [12]). Despite the complicated Fermi surface, the authors were able to identify the presence of an embedded 2D electron subsystem. The observed extreme magnetoresistance originates from the narrowness of the corresponding open orbit, which agrees with the theory by Lifshitz, Azbel, and Kaganov.

The manuscript is well-written, technically sound, uses well-established methods (I cannot comment on the validity of experimental data and sample preparation, however), and addresses an interesting problem. On the other hand, because of the previous findings related to extreme magnetoresistivity, it is not clear how significant are the obtained results (see also a detailed concern below). In addition, the manuscript lacks analytical analysis beyond citing qualitative arguments from Ref.[12].

Thus, before this manuscript can be recommended for publication in Nature Communications, the authors should address the following concerns/suggestions:

1. The manuscript will strongly benefit from a more detailed analytical analysis. I suggest adding a section to the supplemental material discussing the behavior of the resistivity tensor with a magnetic field and, perhaps, temperature. The latter can be important since it governs the deviation from the $n_e = n_h$ limit. In addition, fittings of the experimental data with phenomenological formulas may be also useful in establishing the connection to the theory by Lifshitz, Azbel, and Kaganov.

Thank you for this comment. We strongly resonate with your expressed interest in strengthening the modeling aspects. While we follow this suggestion happily in the revised manuscript, we would like to briefly outline the rationale for not doing so in the first place. The LAK argument is entirely topological, and there is a topological phase transition in orbit space between a material with only localized orbits and one with orbits spanning the entire reciprocal space. This leads to a sudden non-vanishing net Fermi velocity integrated over one period. In turn, the Boltzmann transport equation is truncated at different order, which then gives this striking difference in the high-field limit of the MR. When $\omega_c \tau \gg 1$, the details of the Fermi surface give only marginal contribution and the topology dominates the high-field response, in particular the XMR. The fact that XMR can be switched by field directions, in the high field limit, is entirely based on topology, while the Fermi surface details merely give the quantitative values.

A direct integration of the Boltzmann equation under magnetic fields on such a complex Fermi surface is not reliably possible. The reason is that the Fermi velocity must be precisely computed and integrated over many revolutions, and this suffers from sharp corners and numerical artifacts on such a Fermi surface.

Instead, we now present a toy model that captures the essential physics. This treats all the closed Fermi surfaces as a background and ignores the higher harmonic components that enter via the complex shape of the Fermi tube that hosts the open orbit. The background charge plays a critical role though, as it must compensate the electron density. We consider a cylindrical tube and offset the relative location of its cross-section in the $k_z = \pi/c$ plane compared to the $k_z = 0$ plane. Even though this model does not reflect the FS details, it reproduces the angle-dependence of the PtSn₄ MR very well with only two free fitting parameters, the k_z offset and the scattering time, τ . This result, not surprisingly, reflects the topological and detail-independent nature of the LAK theory.

We resonate with the reviewer that such a concrete model that displays this phenomenon is a valuable addition to the manuscript.

2. In the introduction, it is stated that the extreme magnetoresistance was also observed in other semimetals such as Cd₃As₂. The latter, however, is a Dirac semimetal. Is there any 2D electron subsystem there that would allow for an open orbit and a similar explanation? If not, how generic is the proposed mechanism for other materials that show the extreme magnetoresistance?

Indeed the mechanism for XMR we promote based on LAK is purely semi-classical and applies generically to all (almost-)compensated semi-metals. This also applies to Cd₃As₂, which is a semi-metal close to its charge neutrality point. However, one must make a key distinction: The existence of the open orbit is not important for the existence of the XMR, it

is important for its demise. The fact that Cd_3As_2 does not have open orbits in any direction implies that it must exhibit XMR for any field direction, as it does. The peculiar and novel aspect about PtSn_4 is that the XMR can be switched off. While this is very important to identify the origin of the XMR, this switchability is special and certainly not generic. We have emphasized this point in the manuscript.

3. Is there any intuitive physical explanation for why the physical behavior of PtSn_4 is that of an almost isotropic, strongly three-dimensional intermetallic metal? This is an interesting observation especially taking into account the layered structure of PtSn_4 .

Thank you for raising this point. It is in addition interesting that the only real quasi-2D electronic subsystem emerges perpendicular to the physical planes, which is also counter-intuitive.

Yet it is also clear that intuition about the electronic structure finds its limits given the enormous complexity of the material. A tight binding description is derived from Pt 5s and 5d as well as Sn 5s and 5p. This is a total of $12 + 4 \cdot 8 = 44$ spinful orbitals with strong spin-orbit effects. Not surprisingly, many strongly warped Fermi surface sheets cover the entire Brillouin zone. It is evident that the velocity operator averaged over all these sheets will not have a strong directionality.

An intuitive answer based in real space is that both Pt and Sn contribute equally to states at the Fermi level and conduction process, and hence their layered arrangement does quantitatively renormalize but not substantially lower the dimensionality of transport in this metal. We now made this point apparent in the revised manuscript.

4. While the authors try to provide a connection to applications, extreme sensitivity to the direction of the magnetic field may negate all potential gains from the extreme magnetoresistance. The authors should either tune down these discussions or strengthen them, and explain how the observed phenomenon may be used.

Please excuse any potential misunderstandings of this comment, it does not appear to us that the paper is overly angled towards applications. In the introduction, we describe the important link between unusual magnetoresistance anomalies and XMR in general for technological applications. As this link is well established and does not concern PtSn_4 directly, we assume the comment here refers to the concluding paragraph, in which we

stated “*The presence of singular open orbits provides a unique possibility to selectively switch XMR on and off, a rather rare materials property that may find interesting applications.*”. It is true that this comment is a bit off-hand, following the logic that unique material properties may generically be uniquely suited for applications, but without any specific goal. Certainly, the temperature and field scale of PtSn₄ renders applications rather unlikely. We have deleted this comment.

5. It is mentioned in the summary that PtSn₄ is “an interesting testbed to probe quasi-one-dimensional electron transport in a compensated electron-hole sea”. However, the main finding originates from a hidden 2D electron subsystem. Do the authors understand quasi-1D in the sense that the extreme magnetoresistance is observed only along a certain crystallographic axis? This statement should be clarified.

Thank you for bringing this point up, this was indeed not well explained in the previous version. The key topological distinction between open and closed orbits, when it comes to transport properties, is their non-vanishing Fermi velocity perpendicular to the magnetic field. Closed orbits in k-space correspond in real space either to closed orbits (that then give rise to Landau level physics), or helical states that only propagate along the magnetic field direction. As open orbits do not consist of k-pairs related by time-reversal symmetry, their in-plane velocity averages over a period to a finite value. So these modes propagate in this direction perpendicular to the magnetic field, unlike closed orbits that are localized perpendicular to the field. We have clarified this statement.

6. The axes in Fig.1a) seem to be wrong and contradict the description in the main text and literature, see also Fig. 1 in E. Mun, H. Ko, G. J. Miller, et al., Phys. Rev. B 85, 035135 (2012). Indeed, while it is stated that “alternating layers of Sn and monoatomic Pt planes [are] stacked along the b-direction”, the layers in the figure are stacked along the c-direction. I suggest to double-check the labeling of the axes along the text and to bring them in agreement with the literature.

Thank you for catching this! We are working on so many layered metals which are, more conventionally, stacked along the c-direction. Out of habit, c and b were swapped mistakenly.

Reviewer #3 (Remarks to the Author):

The manuscript by Diaz et al. reports an angle-dependent magnetoresistance study on microstructured devices of high purity metal PtSn₄. The key finding is an abrupt disappearance of the extreme magnetoresistance (XMR) in the system with magnetic fields along b, and the authors highlighted the similarity of the observation with that known in compensated metal zinc, and used this to support that the XMR in the system is likely semiclassical and attributed to compensation of the high mobility carriers. The experimental observation and conclusion themselves are robust and the sharpness of the observation is uniquely enabled by fabrication of mesoscopic devices. However, the novelty and significance of the study, beyond elucidating the mechanism of XMR in the system, remain somewhat unclear. Similar physics and phenomenology of compensated metals have been

known to elemental Zn, and more recently reported in high mobility ReO₃ with comparable magnitude of magnetoresistance (see e.g. Phys. Rev. Materials. 5, 105004, 2021).

Thank you for your positive feedback and for bringing the point of other classical materials up. We have addressed this point in the overarching comment up top, and of course added it to the manuscript.

On top of this, one aspect of the manuscript that needs correction is the labeling of the crystal structure as well as crystallographic orientations. Fig.1a displays a crystal structure seemingly containing more Pt atoms than Sn, contradicting the chemical formula. The choice of a, b and c axes in Fig. 1a is inconsistent with most literature on PtSn₄ (e.g. PhysRevB.103.085125 Fig. 1d) and nor is it consistent with the cell parameters cited in page 2 paragraph 3. These conflicting conventions are again used in a mixed manner in the discussion of the DFT Fermi surface and cyclotron orbits. Given the key importance of the relative orientation of magnetic field to the crystallographic axes in the study, it is important to use a notation that does not cause confusion.

Thank you for pointing this out, in resonance with referee 1. This has been a constant struggle throughout this project, as it is customary in layered conductors to call the q2D direction the *c*-direction. This can be confusing, and we fear this has caused confusion also in this comment. In fact, the only mistake was in Figure 1a, the DFT treatment is correct. This is rooted the interesting yet further confusing finding that PtSn₄ is doubly unconventional. First, the layers stack along the *b*-direction, and second the q2D system is not confined to these layers but exists perpendicular to it. In effect, this means the cylindrical q2D pocket in fact does follow the *c*-direction, as shown correctly in Fig.3. We have fixed the mistake in Fig. 1 of course.

Fig. 1a also had a color mistake in the atomic labels which indeed led to a wrong atomic count.

Below are a few additional questions:

(a)How robust is the narrowly connected Fermi surface feature in DFT (Fig. 4c) with respect to shifts in Fermi energy or relative band positions, which could occur in real materials?

This is in fact quite robust due to the light electronic masses. The narrowest part of this cylinder falls on the *Y* – *F*₀ line. One finds a Lifshitz transition disconnecting this neck at 200meV above the Fermi level, which is an unreasonably high shift to originate from doping given the high electron density and large zero-field conductivity of these crystals.

Hence while these connections may appear small compared to the Brillouin zone, the connections are robust.

(b) Does the highlighted quantum oscillation in Fig. 3 correspond to any of the Fermi surfaces shown in Fig. 4? Are there any quantum oscillation evidences for the evolution between closed and open cyclotron orbits?

The brief answer, sadly, is not conclusively. In more detail, there are multiple reasons for why this is the case and also why we did not pursue this direction. First, the Fermi surface is extremely complex, and matches between DFT and quantum oscillations are very difficult. There are so many frequencies predicted and observed, that matches are almost guaranteed. Some careful work has been done (e.g. T. Yara et al., Physica B 536, 625 (2018)), and still the insights are limited. One can certainly “mix and match” to obtain a reasonable match of the angle dispersion; however there are major orbits unlabeled as well as many theoretically ones unobserved (If you look at the Yara paper, please note that the relevant Fig. 5a does not show theory but theory inspired guide-to-the-eye; while there are substantial discrepancies to the actual theory in 5b). This is not a surprise or shortcoming of this fine work, given the geometric complexity a tomographic reconstruction by quantum oscillations is a challenge.

Second, given the high frequency components of a Fourier decomposition of this complex FS, one needs extreme absolute precision in the determination of both (!) field angles. This is not always easy to achieve. Our frequencies are in good agreement with the published theory, but we could not add significant new insights on the Fermiology from QO here.

Lastly, the open orbit is extremely curved. The QO amplitude within the LK formalism is proportional to the inverse average curvature, hence this orbit and its angular evolution towards the open orbit is expected to suffer from strong geometric damping. Accordingly, we do not see any oscillations that come even close to a match and we are not aware of any publication that claimed even circumstantial matches to this orbit. Of course, if it was observed, the amplitude would be strongly suppressed further in the most interesting region of the transition to the open orbit, which by definition does not show QO.

(c) P7 paragraph 1, “Gamma-X” line is mentioned, but no “X” point can be found in Fig. 4.

We corrected that typo.

(d) What’s the orientation of current in the bulk curve in Fig. 1c?

It is a platelike crystal aligned with the a -direction, in good agreement with the microstructure data. However, we did not want to overemphasize this as in this highly conductive crystal the current lines connecting the contacts may be bend substantially depending on the uncontrolled microscopics of the handmade contacts. The result, however, suggests that this has worked out well and we have added a comment on this.

Reviewers' Comments:

Reviewer #1:

Remarks to the Author:

In the revised version of the manuscript, the authors have addressed my previous concerns and corrected a few mistakes. I appreciate the inclusion of the analytical model and believe that it clarifies the observed strong dependence of the extreme magnetoresistance (XMR) on the direction of a magnetic field. While the absence of "new and exotic physics" may seem like a drawback, it is exciting that new and unusual effects captured by the semiclassical theories can be still experimentally found. The nontrivial experimental execution with fine control over the direction of the magnetic field and precise fabrication of the sample is undoubtedly a strong side of this manuscript.

Since the manuscript combines an experimental observation of the XMR with numerical calculations and a reasonable semiclassical model, I believe that it provides a sufficient contribution to our understanding of XMR. Furthermore, the manuscript demonstrates an unusual and, to the best of my knowledge, previously unobserved switchability of the XMR by tuning the direction of a magnetic field. Such an observation may excite interest in a more refined study of other materials supporting XMR. I recommend this manuscript for publication in Nature Communications.

Reviewer #3:

Remarks to the Author:

The authors have satisfactorily addressed my concerns raised in the previous round. The novelty of this work with respect to relevant studies is sufficiently clarified in the revised manuscript. The demonstration of suppression of XMR in PtSn₄ with the emergence of an open orbit contrasts with the well-established notion that non-saturating (saturating) MR in metals with a dominant carrier is associated with open (closed) orbits, and therefore provides strong evidence that carrier compensation, along with closed orbits are the key to XMR in many recently studied, high mobility semimetals. The present manuscript will attract readers interested in topology, transport in quantum materials and fits the scope of Nature Communications.